# A Transcriptome-Wide Analysis of Psoriasis: Identifying the Potential Causal Genes and Drug Candidates

**DOI:** 10.3390/ijms241411717

**Published:** 2023-07-20

**Authors:** Yeonbin Jeong, Jaeseung Song, Yubin Lee, Eunyoung Choi, Youngtae Won, Byunghyuk Kim, Wonhee Jang

**Affiliations:** 1Department of Life Sciences, Dongguk University, Seoul 04620, Republic of Korea; ybjeong13@gmail.com (Y.J.); jaeseung6455@gmail.com (J.S.); leeyb9916@gmail.com (Y.L.); qhrdma99@gmail.com (E.C.); happywing95@gmail.com (Y.W.); 2Department of Life Sciences, Dongguk University-Seoul, Goyang 10326, Republic of Korea; bkim12@dongguk.edu

**Keywords:** psoriasis, transcriptome-wide association study (TWAS), colocalization, protein–protein network, drug candidates

## Abstract

Psoriasis is a chronic inflammatory skin disease characterized by cutaneous eruptions and pruritus. Because the genetic backgrounds of psoriasis are only partially revealed, an integrative and rigorous study is necessary. We conducted a transcriptome-wide association study (TWAS) with the new Genotype-Tissue Expression version 8 reference panels, including some tissue and multi-tissue panels that were not used previously. We performed tissue-specific heritability analyses on genome-wide association study data to prioritize the tissue panels for TWAS analysis. TWAS and colocalization (COLOC) analyses were performed with eight tissues from the single-tissue panels and the multi-tissue panels of context-specific genetics (CONTENT) to increase tissue specificity and statistical power. From TWAS, we identified the significant associations of 101 genes in the single-tissue panels and 64 genes in the multi-tissue panels, of which 26 genes were replicated in the COLOC. Functional annotation and network analyses identified that the genes were associated with psoriasis and/or immune responses. We also suggested drug candidates that interact with jointly significant genes through a conditional and joint analysis. Together, our findings may contribute to revealing the underlying genetic mechanisms and provide new insights into treatments for psoriasis.

## 1. Introduction

Psoriasis is an immune-related disease that is accompanied by chronic inflammation of the skin [1,2]. Psoriasis affects approximately 2–4% of the global population, and the number is increasing [3,4,5]. Psoriasis is characterized by itchiness, soreness, rashes, and pain in skin lesions [6,7]. Although its etiology has not been clearly determined, multiple factors, including infection, the external environment, and genetic backgrounds, are expected to play important roles in the pathogenesis [6,8,9]. Therefore, an integrative approach is required to identify putative therapeutic targets and druggable molecules for psoriasis prevention and treatment.

Several studies have been conducted over the past decades to identify the psoriasis-causal genes [10,11,12,13]. Genes encoding late cornified envelope (LCE) proteins that function as barriers have been suggested as risk factors for psoriasis in population-level genome-wide association studies (GWAS) and functional studies [12,14,15]. In addition, the genes encoding proteins involved in the nuclear factor κB (NFκB) signaling pathway, including *interleukin 12B* (*IL12B*), *IL23A*, and *tyrosine kinase 2* (*TYK2*), have also been reported as susceptibility genes for psoriasis [16,17,18].

GWAS has the advantage of using large-scale data compared with previous methods, but there is a limitation in interpreting the biological functions of variants existing in non-coding regions. To address this issue, a transcriptome-wide association study (TWAS) using gene expression imputation has been suggested [19]. TWAS predicts the level of gene expression in the phenotype by calculating the association with the genotype using the expression quantitative trait loci (eQTL) panels. Recently, multi-tissue eQTL panels, including context-specific genetics (CONTENT) models and a unified test for molecular signatures (UTMOST), combining tissue-shared genetic features in gene expression regulation, have shown enhanced statistical power in TWAS [20,21]. Considering that TWAS has successfully provided new insights into the pathogenesis of various diseases and prioritized causal genes, it is one of the cutting-edge approaches for investigating the triangulated mechanisms among genetic variants, gene expressions, and phenotypes [22,23,24,25].

Herein, we conducted TWAS for psoriasis using the publicly accessible GWAS summary statistics data from European ancestry. GWAS summary statistics data from the GWAS Catalog (GCST90014456) was used to identify tissues associated with psoriasis using the linkage disequilibrium score-specifically expressed genes (LDSC-SEG) analysis [26,27]. With respect to the tissue prioritization result from LDSC-SEG, we selected eight representative tissue panels (whole blood, sun-exposed skin, not-sun-exposed skin, spleen, transformed fibroblasts, EBV-transformed lymphocytes, esophagus mucosa, and stomach) in Genotype-Tissue Expression (GTEx) version 8. By estimating the gene expression changes driven by genetic variants using TWAS, colocalization (COLOC), and CONTENT analyses, 133 significantly associated genes with psoriasis were identified. To verify robust psoriasis markers, we performed a conditional and joint analysis as a downstream analysis. Functional annotation and network analyses were conducted to interpret the biological mechanisms of psoriasis. Finally, drug candidates that can be used as the treatment options for psoriasis were derived by confirming the gene–drug interactions. The overall workflow of our study is depicted in Figure 1.

## 2. Results

### 2.1. Prioritizing Genetically Relevant Tissue for Psoriasis

TWAS enables the identification of gene-trait associations; however, appropriate tissue selection is essential for obtaining accurate results [28]. To select and focus on the tissues that are most related to psoriasis, we conducted the LDSC-SEG using a multi-tissue RNA expression dataset and a multi-tissue chromatin modification (DNase hypersensitivity, histone acetylation, and histone methylation) dataset [27]. The LDSC-SEG was developed by Finucane et al. as a tool for identifying disease-relevant tissues [27]. They combined the GTEx dataset [29] and the Franke group’s dataset [30,31], which were classified into nine major categories (adipose, blood/immune, cardiovascular, central nervous system, digestive, liver, musculoskeletal/connective, pancreas, and “other”) that could be distinguished as multi-tissue RNA expression. Furthermore, this analysis using a multi-tissue RNA gene expression dataset as a reference LD score suggests that three (blood/immune, digestive, and “other”) out of nine categories have significant associations with psoriasis (false discovery rate (FDR) < 0.05) (Figure 2). Because chromatin modification affects RNA expression, additional analyses were performed using the chromatin modification dataset to verify the results. Similar results were obtained when using the multi-tissue chromatin modification dataset as a reference LD score (Appendix A). Female-specific tissues, such as the vulva and cervix uteri, were identified as being significantly relevant to psoriasis, while male-specific tissues were not. Because it was reported that the prevalence of psoriasis was not gender-specific, we excluded female-specific tissues to prevent undesirable gender bias [32,33,34]. Combining these results, we finally selected eight tissue panels (whole blood, sun-exposed skin, not-sun-exposed skin, spleen, transformed fibroblasts, EBV-transformed lymphocytes, esophagus mucosa, and stomach) from the three categories and the GTEx version 8 for subsequent analyses.

### 2.2. Transcriptome-Wide Associations for Psoriasis

We conducted TWAS using functional summary-based imputation (FUSION) to identify susceptibility genes for psoriasis. FUSION is designed to identify associations between GWAS phenotypes and gene expression values [19]. We used GWAS summary statistics data (GCST90014456), including 329,533 European ancestry (5459 psoriasis patients and 324,074 healthy subjects), and eight tissue panels selected from the GTEx version 8. Among a total of 60,579 associations, 101 genes in 44 loci were significantly associated after applying the multiple testing correction with FDR (FDR < 0.05) (Figure 3A).

The tissue panels with the highest and lowest number of significantly associated genes were sun-exposed skin and EBV-transformed lymphocytes with 35 and seven genes, respectively. In other tissue panels, relatively moderate numbers of genes were significantly associated: 28 in not-sun-exposed skin and esophagus mucosa, 25 in stomach, 23 in whole blood and transformed fibroblasts, and 16 in spleen. A long non-coding *RNA RP11-977G19.11* and a well-known psoriasis risk gene, *interferon regulatory factor 5* (*IRF5*), were identified in every tissue except for spleen and EBV-transformed lymphocytes, respectively.

In order to verify robust genetic markers for psoriasis, we then compared the TWAS results with the COLOC method, which is another gene prioritization method based on the Bayesian test. It calculates the posterior probability (PP) of Hypotheses 0–4 (H0–H4) corresponding to the colocalization patterns of GWAS and eQTL signals, as described in Section 4. We found 130 significant associations with 60 genes (PP3 + PP4 > 0.8 and PP4/PP3 > 2) in eight different tissues, and the majority (81%) of them overlapped with the TWAS signals (Figure 3B).

To increase the statistical power, additional TWAS analyses were performed utilizing the CONTENT panel. The CONTENT panel, designed by Thompson et al., combines and integrates tissue-shared and tissue-specific associations [20]. Consequently, 64 genes were identified (FDR < 0.05) using the CONTENT panels (Appendix A). As a result of the three gene prioritization methods (TWAS, COLOC, and CONTENT), a total of 133 genes were identified, 26 of which were robust genetic markers identified in all three methods (Figure 3C, Appendix A). We also identified five novel psoriasis risk genes, *methionine sulfoxide reductase A*, *elongation factor for RNA polymerase II* (*ELL*), *Myotubularin Related Protein 9* (*MTMR9*), *leucine-rich repeat containing 25* (*LRRC25*), and *single-stranded-DNA-binding protein 4* (*SSBP4*), among 26 genes. At the single-cell level of TWAS using the panels from the California Lupus Epidemiology Study (CLUE) consortium, there was no significant association that passed the multiple testing correction threshold (FDR < 0.05); however, *MTMR9* and *killer-cell lectin-like receptor C4* (*KLRC4*) showed marginal significance (0.05 < FDR < 0.1) in CD8+ T-cell and natural killer cell (NK cell), respectively (Appendix A). These results show that *MTMR9*, one of the novel psoriasis risk genes, may be associated with CD8+ T cells, which are known to be a key cell type in psoriasis [35,36].

### 2.3. Biological Enrichment of Genetic Signatures of Psoriasis

Functional annotation confirmed the biological mechanisms of the 133 previously identified genes. Based on their TWAS-Z-score, genes were categorized as up- or down-regulated genes. Four genes (*methyl-CpG binding domain protein 2*, *REL proto-oncogene* (*REL*), *LCE3D*, and *ring finger protein 145*) were either up- or down-regulated depending on the tissues. Due to the importance of direction of regulation, we excluded these four genes from the list for this analysis only and grouped 129 genes into 74 up-regulated and 55 down-regulated genes. The IL-23-mediated signaling pathway was the most significantly enriched Gene Ontology Biological Process (GOBP) term with the up-regulated genes, consistent with previous studies that IL-23 is a cytokine that plays an essential role in the onset and progression of psoriasis (Figure 4A) [37,38]. Because psoriasis is an immune-related disease, the cytotoxicity and differentiation of lymphocytes including NK cells and T cells were also enriched (Figure 4A). A significant association with lymphocyte cytotoxicity and differentiation was also identified in the Kyoto Encyclopedia of Genes and Genomes (KEGG) pathway (Appendix A). Similarly, down-regulated genes were highly enriched in immune-related and viral-related terms (Figure 4B).

To identify tissue-specific or cross-tissue biological enrichments in psoriasis, functional annotation for each tissue was performed. Among 348 statistically significant pathways, 113 pathways were enriched in multiple tissue panels, indicating that the genetic signature of psoriasis is conserved across tissue panels, showing cross-tissue effects. Figure 4C shows that 10 GOBP enrichments were significantly associated with at least four tissues. In the case of EBV-transformed lymphocytes, a statistically significant result could not be obtained due to an insufficient number of genes. As previous studies suggested that skin and gut microbiota may be one of the causes of psoriasis, the responses to bacterial muramyl dipeptide and peptidoglycan, were significantly enriched in all tissues [39,40]. The production of IL-12, which plays an important role in the pathogenesis of psoriasis, showed a very strong association with two types of skin tissues (not-sun-exposed skin and sun-exposed skin) and stomach [41]. Whole blood and spleen panels showed significant associations in all enrichments. Functional annotation using the KEGG pathway identified necroptosis and/or immune-related pathways associated with psoriasis in every tissue, except for EBV-transformed lymphocytes (Appendix A).

Additionally, a phenome-wide association study (PheWAS) was conducted to check the pleiotropic effects of psoriasis-related genetic features. We identified phenotypes associated with 56 variants from 133 significant psoriasis genes using GWAS ATLAS and 21 phenotype domains passed the Bonferroni-corrected significance threshold (*p*-value < 1.05 × 10^−5^) (Appendix A) [42]. In previous studies, the two most significantly identified domains, immunological and metabolic, have been reported to be strongly associated with psoriasis [43,44,45,46]. Moreover, previous studies have shown that psoriasis patients are also linked to skeletal, psychiatric, and gastrointestinal diseases, which were the third, fourth, and fifth significantly associated domains [47,48,49]. Because most phenotypes from the PheWAS results were already known to be highly associated with psoriasis, our results reflected the general genetic landscape of psoriasis patients hardly being affected by rare cases of comorbidity.

### 2.4. Conditional and Joint Analysis of TWAS Signals

To rigorously assess the significance of TWAS signals from potential inflation by LD contamination, a conditional and joint analysis was conducted on all TWAS significant loci in eight tissues. Because the significance of genes is more important than the direction of regulation, all 133 genes identified in Figure 3C were used in subsequent analyses. After removing expected gene expressions, 75 of the 133 significant genes remained jointly significant. In the case of *IRF5*, which was previously identified in the seven tissue panels (whole blood, sun-exposed skin, not-sun-exposed skin, spleen, transformed fibroblasts, esophagus mucosa, and stomach) in TWAS, the same seven tissues remained statistically significant after the conditional and joint analysis. Meanwhile, for *RP11-977G19.11*, which was previously identified in the seven tissue panels (whole blood, sun-exposed skin, not-sun-exposed skin, EBV-transformed lymphocytes, transformed fibroblasts, esophagus mucosa, and stomach) in TWAS, only two tissues (EBV-transformed lymphocytes and stomach) were jointly significant. In addition, the number of genes for each tissue was relatively reduced after conditioning in all tissues (Appendix A).

Among the five novel genes, three genes (*ELL*, *LRRC25*, and *SSBP4*) remained jointly significant after the conditional and joint analysis (Figure 5, Table 1). While these three genes were located at the same genomic locus (19p13.11), they showed different association patterns across the tissues, which is called the tissue-specific regulatory effect. There are two types of tissue-specific regulatory effects: tissue-specific and tissue-sharing effects. Tissue-specific effects indicate that genes are associated with a specific tissue, while tissue-sharing effects indicate that genes are simultaneously associated with multiple tissues. Additionally, we verified the potential positional effect of eQTLs of the three genes. Utilizing the eQTL browser in the GTEx portal (https://gtexportal.org/ accessed on 28 June 2023), we found that up to 670 potential eQTLs for the three genes reside at the 19p.13.11 locus of each tissue (Appendix A). Among them, less than half of the SNPs showed the statistically significant effect as the eQTL, of which, by filtering them via conditional and joint analysis, only seven SNPs were identified to mediate the tissue-specific gene expression regulation associated with psoriasis (Appendix A).

To increase the reliability of the results, the tissues where three genes were jointly significant were compared with the tissues where the three genes were mainly expressed in a consensus dataset in the Human Protein Atlas (HPA) that integrated HPA RNA-seq data and GTEx RNA-seq data (Appendix A) [50]. First, spleen-specific *LRRC25* also showed the highest expression in HPA spleen and *ELL,* that was specific in whole blood, showed the second-highest expression in HPA bone marrow following testis. However, *SSBP4*, which was jointly significant in the five tissues (sun-exposed skin, not-sun-exposed skin, transformed fibroblasts, esophagus mucosa, and stomach) showed relatively low tissue specificity in the HPA dataset. These results suggest that the expression patterns of *LRRC25* and *ELL* demonstrated tissue-specific effects and those of *SSBP4* showed tissue-sharing effects of psoriasis.

### 2.5. Protein–Protein Interaction Network Analysis and Cluster Identification

We constructed an initial protein–protein interaction (PPI) network using the search tool for the retrieval of interacting genes/proteins (STRING) (ver. 11.5) to investigate how the 133 significant psoriasis genes identified by TWAS, COLOC, and CONTENT analyses (Figure 3C) were systematically connected. We removed genes that were not linked to other genes and networks with three or fewer nodes. Then, three networks made up of 51 genes remained, and the result was visualized. Additional clustering analysis was performed using the molecular complex detection (MCODE) plug-in that identifies highly interacting modules in PPI networks, and four distinctive color-coded sub-clusters were identified (Figure 6). As shown in Figure 6 by the red-lined rhombi, more than half of the genes across the network were jointly significant genes from the conditional and joint analysis results.

The sub-cluster 1, highlighted in orange, consisted of LCE1F, LCE2A, LCE3A, LCE3C, LCE3D, LCE3E, LCE4A, LCE5A, and small proline-rich protein 2D (SPRR2D) that were involved in the keratinization. Previous studies have shown that the LCE family and SPRR2D are involved in the pathogenesis of psoriasis by forming a tough structure beneath the cell membrane during the differentiation of keratinocytes [51,52]. The sub-cluster 2, highlighted in green, was composed of endoplasmic reticulum aminopeptidase 1 (ERAP1), TYK2, tumor necrosis factor receptor-associated factor 3 interacting protein 2 (TRAF3IP2), caspase recruitment domain family member 14, and ring finger protein 114, which are involved in the immune response pathways, including the Janus kinase-signal transducer and activator of transcription (JAK-STAT) and NFκB signaling pathways. The JAK-STAT and the NFκB signaling pathways are the well-known pathways associated with the pathogenesis of psoriasis and are mainly used as therapeutic targets [53,54]. The yellow sub-cluster 3 (ELL, SSBP4, and LRRC25) composed of three novel genes is located at the genomic locus 19p.13.11, and the purple sub-cluster 4 containing signal recognition particle 54, ribosomal protein S26, and chromosome 18 open reading frame 32 is related to the formation of the 40S subunit.

### 2.6. Potential Drug/Chemical Compound Candidates for Psoriasis

To derive drug candidates for psoriasis treatment, gene–drug interaction analysis was performed. Using 75 jointly significant genes as a query set, we fed the data into the Drug Gene Interaction (DGI) database and obtained 268 gene–drug interactions that scored greater than the interaction score of 0. Essentially, only a total of eight genes (*KLRC1*, *TYK2*, *galactokinase 1* (*GALK1*), *N-sulfoglucosamine sulfohydrolase*, *TRAF3IP2*, *ERAP1*, *IL23A*, and *DNA polymerase iota* (*POLΙ*)) among 75 jointly significant genes had overlapping interactions with 261 drugs, establishing 268 gene–drug interactions. Seven genes, excluding *POLΙ*, were included in the top 10% of interactions based on the descending order of interaction scores (Table 2). The highest interaction score was obtained between monalizumab and *KLRC1*. Monalizumab is known to inhibit NKG2A, a receptor protein encoded by *KLRC1*, whose expression is increased in lymphocytes of psoriasis patients [55,56,57]. Drugs that interact with *TYK2* or *IL23A* were JAK-STAT inhibitors or anti-inflammatory monoclonal antibody drugs targeting TYK2 or IL23A, respectively.

## 3. Discussion

TWAS is used to investigate the genetic effects of pathogenesis in various diseases. It has the advantage of estimating the genotype-mediated gene expression changes at the population level as it calculates expected gene expression values using large-scale GWAS summary statistics data [58,59,60,61]. In the GWAS summary statistics data we used, healthy subjects (n = 324,074) were approximately 60 times larger than psoriasis patients (n = 5459), which could cause biased results. Therefore, we calculated the effective sample size, and it seemed unlikely that case to control ratio biased the results. because the genes can show distinct expression patterns in different tissues, it is important to select the correct tissue for tissue-specific analysis [62]. We performed a tissue prioritization process using LDSC-SEG analysis using a multi-tissue RNA expression dataset and a multi-tissue chromatin modification dataset (Figure 2 and Appendix A). Our tissue prioritization results led us to include additional relevant tissues (spleen, EBV-transformed lymphocytes, esophagus mucosa, and stomach) that were not used in the previous psoriasis TWAS study [63]. Previous studies have demonstrated that the spleen, lymphocytes, and gastrointestinal tract are associated with psoriasis [49,64,65,66,67,68]. Some of the marker genes identified in this study may partially be due to the addition of these newly added tissue panels. Using multi-tissue panels from the CONTENT, tissue-specific and tissue-shared effects were also considered. It is very important to identify tissue-specific effects in studying the pathogenesis of diseases; however, most genetic effects are shared across many different tissues [69]. Especially, the TWAS results using two skin tissues (sun-exposed skin and not-sun-exposed skin) showed high similarity (Pearson’s R = 0.88, *p* < 2.2 × 10^−16^). Therefore, it is necessary to integrate both tissue-specific and tissue-shared effects to identify robust genetic markers of diseases.

In addition to utilizing eight different tissue panels that are highly associated with psoriasis, we performed three different gene prioritization methods (TWAS, COLOC, and CONTENT). We believed that integrating the three approaches could complement the deficiencies of each method and identified 133 significant genes for psoriasis (Figure 3). To investigate the biological mechanisms of 133 significant genes, we grouped them into up- and down-regulated genes and performed functional annotation (Figure 4). The genes were enriched in signaling pathways related to cytokines, including interleukins and interferons, which properly conformed with well-known pathogenic mechanisms of psoriasis [70,71,72,73]. Cytokines are also known to have significant effects on the severity of psoriasis lesions [70]. In tissue-specific enrichment analysis (Appendix A), necroptosis was significantly identified in both sun-exposed and not-sun-exposed skin tissues. Necroptosis is a regulated inflammatory mode of cell death that exhibits both aspects of necrosis and apoptosis [74]. Necroptosis, programmed necrosis, is mediated by several cytokines and receptor-interacting protein kinase 1 regulated by *SPATA2*, which is one of the significant genes identified in all tissue prioritization analyses (Appendix A) [75,76,77]. Previously, several studies elucidated the association between necroptosis and psoriasis, showing that inhibiting keratinocyte necroptosis can be an effective treatment strategy for psoriatic inflammation [78,79,80].

As mentioned in Section 2, conditional and joint analysis is required to identify rigorous causal genes by removing potential LD contamination-induced inflation. In this study, we identified three novel genes associated with psoriasis after the conditional and joint analysis (Figure 5 and Table 1): *LRRC25*, *SSBP4*, and *ELL*. We also identified seven eQTLs showing statistically significant effects on the regulation of the tissue-specific gene expression of these three genes (Appendix A). Previous studies have shown that LRRC25 inhibits the NFκB signaling pathway and inflammatory responses by promoting the degradation of the NFκB p65 subunit [81,82,83]. It is also known that the expression of *LRRC25* is regulated by vitamin D, which reduces the progression of autoimmune diseases [84,85]. Because psoriasis patients in previous studies tended to have low vitamin D levels, increasing vitamin D levels through consumption or synthesis by appropriate sun exposure may promote the expression of *LRRC25* and alleviate inflammatory responses [86,87]. Next, SSBP4 binds to the transcriptional activation domain of *interleukin 36 receptor antagonist* (*IL36RA*) and affects the activation of IL-36RA [88,89,90]. IL-36RA deactivates IL-1 and IL-36, which are present in high levels in psoriasis patients [91,92]. Therefore, down-regulation of *SSBP4* may lead to less activation of IL-36RA and cause and/or worsen psoriasis. Finally, ELL binds to the genes involved with the proliferation of keratinocytes and stabilizes RNA polymerase II to sustain cell proliferation [93]. Because one of the main characteristics of psoriasis patients is keratinocyte proliferation, up-regulation of *ELL* can cause psoriasis [94]. These genes were located at the same genomic locus 19p.13.11 and were identified as a single sub-cluster in PPI (Figure 6).

We identified the interactions between genes and drug/chemical compounds to derive potential drug candidates for psoriasis (Table 2). There are drugs already in use for the treatment or under study for the treatment of psoriasis/psoriatic arthritis, including tofacitinib, peficitinib, and solcitinib [53,95,96]. These drugs are JAK-STAT inhibitors targeting TYK2, which belongs to the JAK family along with JAK1, JAK2, and JAK3 [97]. JAK-STAT inhibitors are used to treat for skin diseases by blocking the function of immune-related pathways [98]. Therefore, JAK-STAT inhibitors such as delgocitinib and oclacitinib, which are not currently used for treating psoriasis, may also be used as therapeutic options for psoriasis after a full-panel toxicological study. Esculetin, which interacts with ERAP1, is extracted from *Fraxinus rhynchophylla* Hance and is used as a herbal medicine in Asian countries (Table 2) [99,100]. It is known to have antioxidant, anti-inflammatory, and anti-apoptotic activities, and in particular, it suppresses the NFκB signaling pathway [101,102,103,104,105]. A study using imiquimod-induced psoriasis-like mouse models showed that esculetin alleviated the severity of skin lesions [106]. Therefore, it may have the potential to alleviate psoriasis symptoms in humans. Although some anti-cancer drugs, including monalizumab, showed the highest interaction score, a very careful toxicological assessment is required to be even considered as a psoriasis treatment option.

Although our study has contributed to comprehending the underlying mechanisms of psoriasis, a couple of limitations exist. Because our study was conducted with only computational analyses, experimental studies are required to validate significant genes. Because we used GWAS summary statistics data consisting only of European ancestry, the results of this study did not consider other races. In addition, we were unable to provide the exact effect sizes at the individual-level for the risk factors identified in our study, because we used population-level summary statistics. Lastly, some of the drug candidates are currently in use or under study treatments for cancer and many anti-cancer drugs are usually toxic to normal cells. Full-panel toxicological studies should be warranted to be considered as treatment options for psoriasis [57,107]. Despite these limitations, we believe that our study may contribute to revealing the underlying mechanisms of psoriasis by utilizing eQTL panels that were not used in previous studies and integrating three different gene prioritization methods (TWAS, COLOC, and CONTENT). In addition, we identified three novel genes after the conditional and joint analysis and provided new insight into treatments for psoriasis by suggesting potential drug candidates.

## 4. Materials and Methods

### 4.1. Data Collection and Pre-Processing

GWAS summary statistics data for TWAS analysis were retrieved from the GWAS Catalog (GCST90014456) [26]. The data consisted of 329,533 Europeans that were made up of 5459 psoriatic patients and 324,074 healthy subjects. Because the Z-score for each SNP was not provided in the original GWAS summary statistics data, the Z-score was calculated as follows:$$Z=\frac{log\;(Odd\;Ratio)}{Standard\;Error}=\frac{Beta}{Standard\;Error}$$

For subsequent analysis, the LDSC software (ver. 1.0.1) was used to convert the GWAS summary statistics data to the LD score format [108]. The LD structure of the 1000 Genome Project was used as a reference LD [109]. Pre-computed eQTL panels from the GTEx version 8 consortium were retrieved from the FUSION website (http://gusevlab.org/projects/fusion/ accessed on 12 April 2022) [19,110].

### 4.2. LDSC-SEG

To prioritize tissue types for TWAS, a tissue-specific heritability enrichment analysis was performed for GWAS summary statistics data using LD score regression applied to the LDSC-SEG method [27]. Multi-tissue gene expression and multi-tissue chromatin modification (DNase hypersensitivity, histone acetylation, and histone methylation) data from the previous study by Finucane et al. were used for the analyses [27]. The tissue panels that passed the threshold (coefficient *p*-value < 0.05) were selected for use in TWAS.

### 4.3. Transcriptome-Wide Association Analysis

The overall analysis pipeline for TWAS followed the contents of the FUSION (http://gusevlab.org/projects/fusion/ accessed on 12 April 2022), and the FDR threshold (FDR < 0.05) was applied as the multiple testing correction method. The predicted gene expression changes were computed with linear-based models for the eight tissues (whole blood, sun-exposed skin, not-sun-exposed skin, spleen, transformed fibroblasts, EBV-transformed lymphocytes, esophagus mucosa, and stomach) that showed significant heritability for GWAS signals in the previous step. The results of the best-performing model for each gene were selected as expected changes in gene expression. To ensure the robustness of TWAS signals, permutation tests using the FUSION were performed (number of permutations: 100,000).

### 4.4. Colocalization

Because the FUSION performs TWAS based on the triangulated association among genotype, gene expression, and phenotype, the associations may contain statistical bias caused by LD contamination. Therefore, we validated the results from the FUSION with Bayesian test-based gene prioritization method, COLOC [111]. There are five hypotheses (H0, H1, H2, H3, and H4) regarding whether the variant has a significant association between the GWAS signal and the eQTL, and the posterior probabilities for each hypothesis are calculated [112,113]. Each hypothesis stands for the following circumstances [112]. H0: there is no causal variant; H1: there are only causal variants between genotype and phenotype; H2: there are only causal variants for eQTL; H3: phenotype and gene expressions are driven by two independent causal variants; and H4: phenotype and gene expressions share one causal variant. We set the threshold for the COLOC as PP3 + PP4 > 0.8 and PP4/PP3 > 2 following previous studies [58,114].

### 4.5. Multi-Tissue Signals Using CONTENT

To obtain additional TWAS associations with increased statistical power, multi-context panels from the CONTENT were utilized [20]. We used the CONTENT (full) panel that showed the best results in the original paper by integrating the CONTENT (tissue-shared) panel and the CONTENT (tissue-specific) panel. Eight tissue panels (whole blood, sun-exposed skin, not-sun-exposed skin, spleen, transformed fibroblasts, EBV-transformed lymphocytes, esophagus mucosa, and stomach) used in single-tissue analysis were utilized to identify multi-tissue signals. Furthermore, single-cell RNA-sequencing data from the CLUE were also used as eQTL panels to examine the associations with psoriasis at the single-cell level.

### 4.6. Functional Annotation of Significant Psoriasis Genes

To identify the functional roles of psoriasis markers that were significantly identified at least once in TWAS, COLOC, and CONTENT, functional annotation analysis was conducted. Enrichr, a web-based tool for biological function or pathway analysis of gene lists, was used for functional annotation [115]. We grouped the gene sets by panels or by up- and down-regulated genes. The groups were subjected to enrichment analysis via GOBP and KEGG pathways [116,117].

### 4.7. Conditional and Joint Analysis

Conditional and joint analysis using the post-process function of the FUSION was performed to identify independent TWAS signals at the same loci in each panel. Genes that passed the FDR threshold (FDR < 0.05) were subjected to conditioning the signals. To identify robust genetic signatures, the *p*-values of TWAS signals were compared before and after conditioning. Genes with a significant *p*-value even after conditioning were defined as jointly significant genes.

### 4.8. Network Analysis with Clustering

To identify the systematic interconnections underlying psoriasis etiology, significant genes identified in TWAS, COLOC, and CONTENT were used as nodes for network analysis. PPI networks were constructed using STRING (https://string-db.org/, accessed on 28 June 2023) (ver. 11.5) with medium confidence (interaction score > 0.4) [118]. The networks were visualized using Cytoscape app (ver. 3.9.1), and only networks consisting of more than three nodes were displayed [119]. Among the networks, the highly interconnected regions (sub-clusters) were identified by MCODE plug-in (ver. 2.0.2) [120].

### 4.9. Gene–Drug Interaction Analysis

Gene–drug interaction analysis was conducted to identify potential drug candidates for psoriasis using the DGI database. The DGI database provides integrated information on gene–drug interactions and druggable genes from other publications, databases, or other web-based sources [121]. Drugs having an interaction score greater than 0 with jointly significant genes were regarded as potential drug candidates for psoriasis.

## Figures and Tables

**Figure 1 ijms-24-11717-f001:**
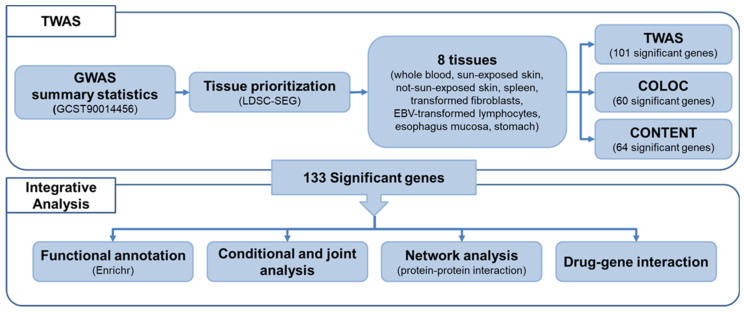
Workflow of the overall study. The data obtained from the GWAS catalog (GCST90014456) were used, and eight tissues were selected through the tissue prioritization process. TWAS, COLOC, and CONTENT analyses were conducted with single- and multi-tissue panels. Downstream analyses were conducted on 133 significant genes.

**Figure 2 ijms-24-11717-f002:**
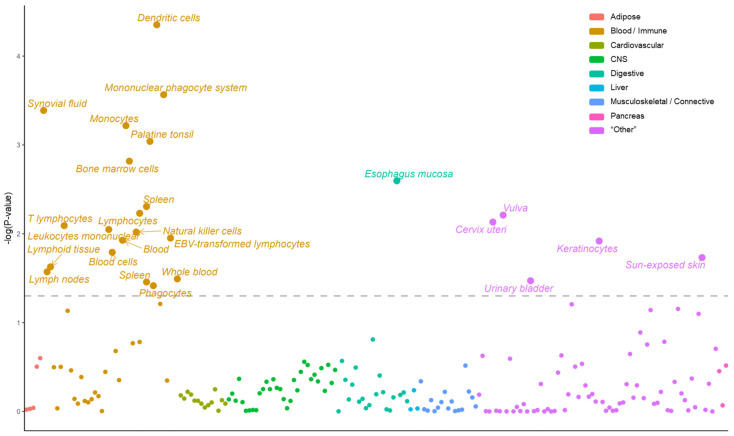
Tissue prioritization using the LDSC-SEG. A scatter plot depicting the tissue prioritization via the LDSC-SEG. GTEx data [29] and Franke group’s data [30,31] were divided into nine big categories (scarlet: adipose; mustard: blood/immune; yellow green: cardiovascular; green: central nervous system; blue green: digestive; turquoise: liver; light blue: musculoskeletal/connective; pink: pancreas; and lilac: “other”) [27]. The *Y*-axis denotes −log(*p*-value) and the gray dotted line indicates cutoff (FDR < 0.05).

**Figure 3 ijms-24-11717-f003:**
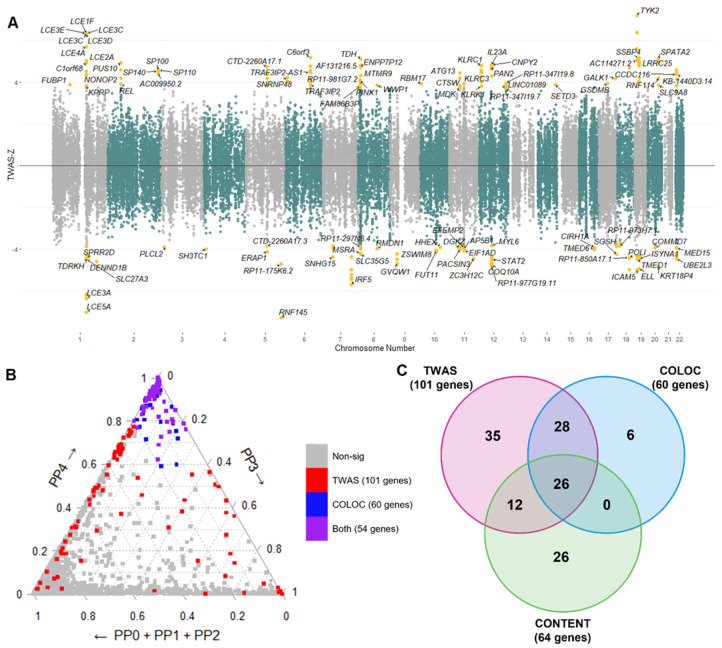
Results of the TWAS and COLOC analyses. (**A**) A Manhattan plot showing the integrative results of TWAS in eight tissues. The *X*-axis indicates the chromosome number where the gene is located, while the *Y*-axis denotes the TWAS-Z-score of the TWAS signals. Genes that passed the cutoff (FDR < 0.05) are highlighted in yellow. If a gene was simultaneously identified in several tissues, the gene is marked with the highest absolute value. (**B**) A ternary plot of the COLOC results. Grey, red, blue, and purple dots represent non-significant genes in any analyses, significantly associated genes in TWAS, significantly colocalized genes in the COLOC, and significantly associated genes in both TWAS and COLOC, respectively. (**C**) A Venn-diagram showing the number of significant genes identified in TWAS, COLOC, and CONTENT.

**Figure 4 ijms-24-11717-f004:**
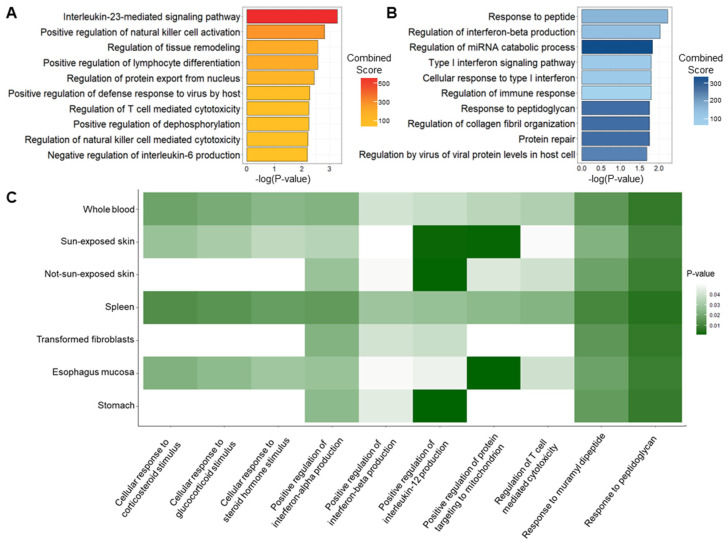
Functional annotation results of significant genes. A bar plot depicting the top 10 enrichments of (**A**) up-regulated and (**B**) down-regulated genes in GOBP. The *X*-axes denote -log(*p*-value) and the color of the bars shows the combined scores from Enrichr. (**C**) A heatmap showing tissue-specific enrichments. The color of each cell represents the *p*-value for each tissue.

**Figure 5 ijms-24-11717-f005:**
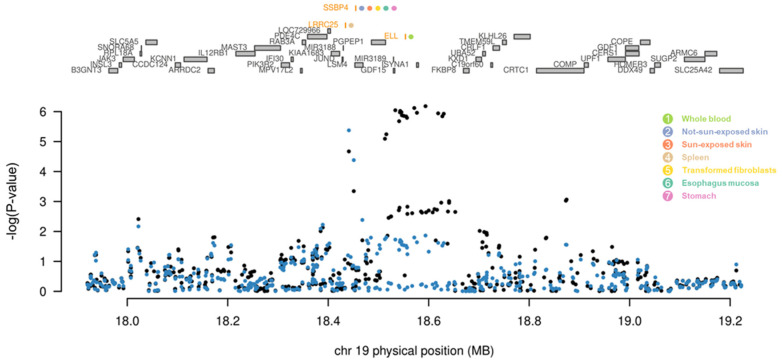
Result of conditional and joint analysis on the 19p13.11 locus. A regional association plot of chromosome 19. Genes in orange colors directly on top of the graph indicate the jointly significant genes that best explain the GWAS signals. Colored dots next to jointly significant genes suggest tissue panels where the gene was identified. Grey bars indicate the location of genes on chromosome 19. The bottom graph shows a Manhattan plot of the GWAS signals. Black and blue dots indicate GWAS-*p*-values before (black) and after (blue) conditioning on jointly significant genes.

**Figure 6 ijms-24-11717-f006:**
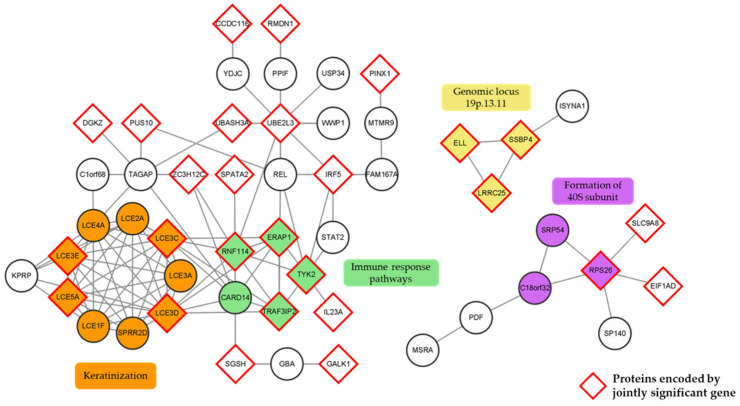
Network analysis using significant psoriasis genes identified via TWAS, COLOC, and CONTENT. PPI was constructed using STRING and visualized using Cytoscape. Only networks with more than three nodes are shown. The color of the nodes indicates the sub-clusters assigned with the MCODE plug-in. Rhombic-shaped nodes with red outlines are jointly significant genes in at least one tissue panel.

**Table 1 ijms-24-11717-t001:** Tissue-specific results of novel genes from conditional and joint analysis.

Gene	Z (TWAS)	P (TWAS)	Z (Joint)	P (Joint)	Tissue	Direction of Regulation
*ELL*	4.3	1.6 × 10^−5^	4.3	1.6 × 10^−5^	Whole blood	Up-regulated
*LRRC25*	−5	5.8 × 10^−7^	−5	5.8 × 10^−7^	Spleen	Down-regulated
*SSBP4*	−4.9	9.5 × 10^−7^	−4.9	9.5 × 10^−7^	Sun-exposed skin	Down-regulated
	−4.8	1.6 × 10^−6^	−4.8	1.6 × 10^−6^	Not-sun-exposed skin	Down-regulated
	−5.0	6.6 × 10^−7^	−5.0	6.6 × 10^−7^	Transformed fibroblasts	Down-regulated
	−5.0	6.4 × 10^−7^	−5.0	6.4 × 10^−7^	Esophagus mucosa	Down-regulated
	−5.2	2.6 × 10^−7^	−5.2	2.6 × 10^−7^	Stomach	Down-regulated

**Table 2 ijms-24-11717-t002:** The top 10% of drug candidates interacting with jointly significant genes.

Gene	Drug Candidate	Interaction Score	Gene	Drug Candidate	Interaction Score
*KLRC1*	Monalizumab	127.30	*TRAF3IP2*	Nevirapine	2.77
*SGSH*	N-sulfoglucosamine sulfohydrolase recombinant	63.65	*GALK1*	Pyrantel pamoate	1.33
				Tricetin	0.66
				Suramin hexasodium	0.48
*IL23A*	Guselkumab	31.83	*TYK2*	Brepocitinib	1.03
	Tildrakizumab	10.61		Tofacitinib	0.91
	Brazikumab	10.61		Peficitinib	0.82
	Risankizumab	10.61		Delgocitinib	0.51
	Briakinumab	7.07		Tofacitinib citrate	0.51
	Ustekinumab	6.37		BMS-911543	0.51
*ERAP1*	Umbelliferone	5.30		Oclacitinib	0.51
	Scopoletin	1.77		Solcitinib	0.51
	Esculetin	1.77		Cerdulatinib	0.46
	Tosedostat	0.56		Upadacitinib	0.34

## Data Availability

Data sharing not applicable.

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
