# Peer review of "A Transcriptome-Wide Analysis of Psoriasis: Identifying the Potential Causal Genes and Drug Candidates"

_ijms, 2023, doi:10.3390/ijms241411717_

Round 1

Reviewer 1 Report

The authors perform a TWAS study for Psoriasis using a dataset previously used for a similar GWAS study. 

1. The dataset has a combination of normal and Psoriasis patient samples. Most of the samples are normal. Do the authors think this influenced/biased their results in any way? Especially the reduced vitamin D levels? 

2. Did the authors attempt any comparisons of sun-exposed and not-sun-exposed skin?

3. Lastly, the TWAS confirms the previously known Psoriasis markers. In terms of over all significance of the study how do the authors reaffirm the importance of their methods?

Reviewer 2 Report

The present manuscript describes an in-silico study aimed at identifying transcriptome signatures associated with psoriasis. Overall, the study seems to be well conducted. However, there are issues to be addressed:

-Authors employed a publicly available dataset. Did authors evaluate the presence of comorbidity in the psoriatic patients (if available)? This issue can influence their results, especially considering the multitissue approach.

- In lines 214-215, authors stated that the identified novel genes showed different association patterns related to possible tissue specificity. I think that authors could add a table/figure or modify the Table 1 for helping the visualization of such data as well as a broad comprehension (for instance, clearly add down or up-regulated and in which tissue). Moreover, the fact that the three genes are located within the same genomic locus 19p.13.11 seem to suggest that there can be various genetic variants in Linkage Disequilibrium (LD) throughout the sequences of three genes, that may have been annotated also as eQTLs. Are there any data in literature? Authors should add a discussion on this subject, considering potential position effect of different genetic variants; namely, how, if existing, variants in LD known to be eQTLs for the three genes may account for different expression patterns. I think that such an integration could help to better explain the interaction of genetic variants (especially considering the utilization of GWAS statistics by the authors) with gene expression profiles.

-In lines 369-370, authors stated that they identified risk factors for psoriasis, including the novel genes. Is this conclusion supported by dedicated analyses, such as an Odd ratio evaluation? The authors should provide an effect size, especially for the differential expression of the new genes, such as Cohen index.

The manuscript only requires a revision for minor language errors and typos.

Round 2

Reviewer 2 Report

Authors revised the manuscript according to reviewer's comments. Overall, upon this revision, the manuscript is improved in term of quality and presentation.

The quality of language is adequate.